

# Lattice Bisognano-Wichmann modular Hamiltonian in critical quantum spin chains

Jiaju Zhang[1,2], Pasquale Calabrese[1,2,3], Marcello Dalmonte[1,3] and M. A. Rajabpour[4]

**1** Scuola Internazionale Superiore di Studi Avanzati (SISSA),
Via Bonomea 265, 34136 Trieste, Italy
**2** INFN Sezione di Trieste, Via Bonomea 265, 34136 Trieste, Italy
**3** International Centre for Theoretical Physics (ICTP),
Strada Costiera 11, 34151 Trieste, Italy
**4** Instituto de Fisica, Universidade Federal Fluminense,
Av. Gal. Milton Tavares de Souza s/n, Gragoatá, 24210-346, Niterói, RJ, Brazil

## Abstract

We carry out a comprehensive comparison between the exact modular Hamiltonian and the lattice version of the Bisognano-Wichmann (BW) one in one-dimensional critical quantum spin chains. As a warm-up, we first illustrate how the trace distance provides a more informative mean of comparison between reduced density matrices when compared to any other Schatten $n$-distance, normalized or not. In particular, as noticed in earlier works, it provides a way to bound other correlation functions in a precise manner, i.e., providing both lower and upper bounds. Additionally, we show that two close reduced density matrices, i.e. with zero trace distance for large sizes, can have very different modular Hamiltonians. This means that, in terms of describing how two states are close to each other, it is more informative to compare their reduced density matrices rather than the corresponding modular Hamiltonians. After setting this framework, we consider the ground states for infinite and periodic XX spin chain and critical Ising chain. We provide robust numerical evidence that the trace distance between the lattice BW reduced density matrix and the exact one goes to zero as $\ell^{-2}$ for large length of the interval $\ell$. This provides strong constraints on the difference between the corresponding entanglement entropies and correlation functions. Our results indicate that discretized BW reduced density matrices reproduce exact entanglement entropies and correlation functions of local operators in the limit of large subsystem sizes. Finally, we show that the BW reduced density matrices fall short of reproducing the exact behavior of the logarithmic emptiness formation probability in the ground state of the XX spin chain.

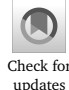

# 1 Introduction

Measures of quantum entanglement and, in particular, entanglement entropies have become one of the key tools to characterize quantum many-body systems and quantum field theories [1–5]. Given a pure state $|\Psi\rangle$ and a subsystem $A$, the bipartite entanglement is quantified by the von Neumann entropy

$$S_A = -\text{tr}_A(\rho_A \log \rho_A),\tag{1}$$

where the reduced density matrix (RDM) $\rho_A = \text{tr}_{\bar{A}}|\Psi\rangle\langle\Psi|$ is obtained by tracing $|\Psi\rangle\langle\Psi|$ over $\bar{A}$ the complement of A. More generally, one can consider the moments of the RDM, i.e. $\text{tr}_A \rho_A^n$, and define the Rényi entropy as

$$S_A^{(n)} = -\frac{\log \text{tr}_A \rho_A^n}{n-1}.\tag{2}$$

In the $n \to 1$ limit, the Rényi entropy returns the entanglement entropy; we note that Rényi entropies of integer order have already been experimentally measured up to partitions consisting of 10 spins [6–11]. The RDM $\rho_A$ is fully encoded in the modular (or entanglement) Hamiltonian $H_A$ defined as

$$\rho_A = \frac{e^{-H_A}}{Z_A}, \qquad Z_A = \text{tr}_A e^{-H_A}.\tag{3}$$

By construction, the RDM and the modular Hamiltonian have the same eigenvectors, and their eigenvalues are simply related. The modular Hamiltonian plays a key role in quantum field theory [5], and recently attracted a great amount of interest in the context of condensed matter physics, in particular, for topological matter [12].

While, in the context of relativistic quantum field theory, the modular Hamiltonian of half-space partition is known to be related to the boost operator [13, 14], its explicit functional

form in lattice models is hard to construct, and is known only in a few simple cases [15–22]. In order to overcome this challenge, it was proposed in Refs. [23, 24] to use the Bisognano-Wichmann (BW) theorem in quantum field theory [13, 14] and its extension in conformal field theory (CFT) [25–34] to write approximate modular Hamiltonians for lattice models. From the BW modular Hamiltonian one can construct a RDM, which has been dubbed BW RDM. The proposal has been checked extensively [20, 23, 24, 35–41], showing that in many cases the BW modular Hamiltonian can reproduce to a good precision the entanglement spectrum, correlation functions, entanglement entropy and Rényi entropies.

In this paper we further investigate the precision of the lattice BW modular Hamiltonian when *the size of the subsystem becomes large*. It was found in [19] that for a free fermion chain the deviation of the BW modular Hamiltonian from the exact one persists even for a large subsystem. However, two RDMs with asymptotically zero distance can have very different modular Hamiltonians, as we will show in section 3 using a toy example, and so the modular Hamiltonian itself may not be a good quantity to distinguish different states of the subsystem. Recently in [41], the Schatten distances, with a normalization as in [42], between the BW RDM and the exact RDM were calculated, and it was found that they decay algebraically with interval's size. In this paper we want to study the same problem using the trace distance. There are at least two good reasons to study also the trace distance between the two reduced density matrices. First, the behaviour of trace distance between the BW and the exact RDMs imposes strong constraints on the behaviour of other quantities such as correlation functions, entanglement entropy, and Rényi entropies through various inequalities. The verification of these operator inequalities was so far essentially neglected. Second, as we will show in section 3, there exist rather different states that the Schatten distance of a large subsystem, normalized or not, cannot distinguish, while the trace distance can.

Since in this article we just use the BW modular Hamiltonian of two-dimensional (2D) CFTs, we will focus on two critical points of the XY spin chain, i.e., the XX spin chain with zero magnetic field and the critical Ising spin chain. The XY chain can be exactly diagonalized and this allows us to calculate the trace distance between the BW and the exact RDMs for one interval with length as large as $\ell \sim 100$. Furthermore, we also calculate the fidelity, which can give an upper bound of the trace distance, for intervals up to size $\ell \sim 1000$ for XX chain and $\ell \sim 500$ for critical Ising chain. We find a power-law decay of the trace distance in $\ell$. Then, exploiting the inequalities satisfied by the trace distance, we will look into the behavior of entanglement entropy, Rényi entropy, RDM moments, correlation functions, and formation probabilities.

The remaining part of the paper is arranged as follows. In section 2 we first define the trace distance and then summarize its relevant properties. In section 3, we use toy examples to show that the trace distance can distinguish states that any other $n$-distance, normalized or not, cannot; we also show that two close RDMs in general can have different modular Hamiltonians. In section 4 and section 5, we consider the ground state of the XX spin chain with a zero transverse field and of the critical Ising spin chain on an infinite straight line and on a circle. We conclude with discussions in section 6.

## 2 Trace distance and its properties

For two density matrices $\rho, \sigma$ with $\mathrm{tr}\rho = \mathrm{tr}\sigma = 1$, the Schatten $n$-distance with $n \geq 1$ is defined as [43, 44]

$$D_n(\rho, \sigma) = \frac{1}{2^{1/n}} \|\rho - \sigma\|_n, \tag{4}$$

where for a general matrix $\Lambda$, Schatten $n$-norm with $n \geq 1$ is

$$\|\Lambda\|_n = \Big( \sum_i \lambda_i^n \Big)^{1/n}, \tag{5}$$

where $\lambda_i$'s are the singular values of $\Lambda$, i.e. the nonvanishing eigenvalues of $\sqrt{\Lambda^\dagger \Lambda}$. When $\Lambda$ is Hermitian, $\lambda_i$'s are the absolute values of the nonvanishing eigenvalues of $\Lambda$. We have chosen the normalization such that for two orthogonal pure states we have $D_n = 1$. The $n$-distance is not always able to distinguish different states, and in [42] an alternative definition of the $n$-distance has been proposed[1]

$$\tilde{D}_n(\rho, \sigma) = \frac{\|\rho - \sigma\|_n}{(\mathrm{tr}\rho^n + \mathrm{tr}\sigma^n)^{1/n}}, \tag{6}$$

which can be called normalized $n$-distance. For $n = 1$, the $n$-distance becomes the trace distance

$$D(\rho, \sigma) = \frac{1}{2} \|\rho - \sigma\|_1. \tag{7}$$

As we will show in section 3, the trace distance can distinguish, in a way that we specify in detail below, some states of a large subsystem that any other $n$-distance, normalized or not, cannot. This is in agreement with the known metric properties of the trace distance (see, e.g., Ref. [47, 48]); it also confirms some observations done for excited states of CFTs [49, 50], as well as out of equilibrium [51].

The fidelity of two density matrices $\rho$ and $\sigma$ are defined as [43, 44][2]

$$F(\rho, \sigma) = \mathrm{tr} \sqrt{\sqrt{\sigma} \rho \sqrt{\sigma}}. \tag{8}$$

Although not obvious by definition, the fidelity is symmetric to its inputs. The fidelity provides both a lower bound and an upper bound on the trace distance

$$1 - F(\rho, \sigma) \leq D(\rho, \sigma) \leq \sqrt{1 - F(\rho, \sigma)^2}. \tag{9}$$

The upper bound will be extremely useful to us.

The trace distance also bounds other interesting quantities such as the entanglement entropy, Rényi entropies, RDM moments, and the correlation functions. The interested reader can find all necessary details in the review [52]. The Fannes-Audenaert inequality provides an upper bound for the entanglement entropy difference [53, 54]

$$|S_A(\rho_A) - S_A(\sigma_A)| \leq D \log(d_A - 1) - D \log D - (1 - D) \log(1 - D), \tag{10}$$

where $D \equiv D(\rho_A, \sigma_A)$ is the trace distance and $d_A$ is the dimension of the RDM. For one interval with $\ell$ sites (e.g. in the XY spin chain), one has $d_A = 2^\ell$. Hence, if the trace distance decays faster than $1/\ell$, then difference of the von Neumann entropies goes to zero for the large subsystem sizes.

The trace distance also puts bounds on the difference of the Rényi entropies and RDM moments. For Rényi entropies with $0 < n < 1$ one has [54]

$$|S_A^{(n)}(\rho_A) - S_A^{(n)}(\sigma_A)| \leq \frac{1}{1-n} \log[(1-D)^n + (d_A - 1)^{1-n} D^n], \tag{11}$$

---

[1]The $n = 2$ version of the normalized $n$-distance (6) was proposed and used in [42] to investigate the time evolution of the RDM after a global quench [45, 46]. As stated in [42], the triangle inequality for the normalized $n$-distance has not been proven yet, and so it may even not be a real well-defined distance.

[2]In quantum information literature, Eq. (8) is sometimes called square root fidelity and the fidelity is defined as $F(\rho, \sigma) = \big(\mathrm{tr}\sqrt{\sqrt{\sigma}\rho\sqrt{\sigma}}\big)^2$.

while for $n > 1$ [55]

$$|S_A^{(n)}(\rho_A) - S_A^{(n)}(\sigma_A)| \le \frac{d_A^{n-1}}{n-1}\Big[1 - (1-D)^n - \frac{D^n}{(d_A-1)^{n-1}}\Big]. \tag{12}$$

According to both inequalities, the trace distance should decay exponentially fast in $\ell$ to get a vanishing upper bound. It is worth mentioning that both inequalities are known to be sharp [55]. We note that since in 2D CFTs we have [56, 57]

$$S_{A,\text{CFT}}^{(n)}(\ell) = \frac{c}{6}\Big(1 + \frac{1}{n}\Big)\log \ell + \gamma_n \tag{13}$$

for critical systems the quantity $\left|1 - S_A^{(n)}(\ell)/S_{A,\text{CFT}}^{(n)}(\ell)\right|$ can be bounded by zero if: a) for $n = 1$ the trace distance goes to zero like $\frac{1}{\ell^\alpha}$ with $\alpha \ge 1$; b) for $n \neq 1$ the trace distance goes to zero exponentially fast.

For RDM moments, there are also bounds. For $0 < n < 1$ one has [54]

$$|\text{tr}_A \rho_A^n - \text{tr}_A \sigma_A^n| \le (1-D)^n + (d_A - 1)^{1-n} D^n - 1, \tag{14}$$

and for $n > 1$ [58]

$$|\text{tr}_A \rho_A^n - \text{tr}_A \sigma_A^n| \le \frac{2D}{n}. \tag{15}$$

The last equation implies that, when the trace distance decays to zero for the large subsystems, the difference of the RDM moments also goes to zero for $n > 1$. The same conclusion is not true for $0 < n < 1$. Anyhow, in order to have a meaningful constraint, we should also take into account how $\text{tr}_A \rho_A^n$ itself scales with $\ell$. For 2D CFTs the moment scales as $\text{tr}_A \rho_{A,\text{CFT}}^n \propto \ell^{-\frac{c}{6}(n-\frac{1}{n})}$ which means the quantity $\left|1 - \frac{\text{tr}_A \rho_A^n}{\text{tr}_A \rho_{A,\text{CFT}}^n}\right|$ can be bounded by zero if: a) for $n < 1$ the trace distance goes to zero exponentially fast; b) for $n > 1$ the trace distance scales like $\ell^{-\alpha}$ with $\alpha > \frac{c}{6}(n-\frac{1}{n})$.

Finally, the trace distance gives the following constraint on the difference of the expectation value of an operator [43]

$$|\text{tr}_A[(\rho_A - \sigma_A)\mathcal{O}]| \le s_{\max}(\mathcal{O})\|\rho_A - \sigma_A\|_1, \tag{16}$$

where $s_{\max}(\mathcal{O})$ is the largest singular value of $\mathcal{O}$. Hence, if the trace distance between two density matrices goes to zero for large subsystem sizes, then the difference between the expectation values of the operators, with finite largest singular value, calculated using the two density matrices goes to zero. Since most of the local operators in quantum spin chains have finite $s_{\max}(\mathcal{O})$, the trace distance puts a strong constraint on their value.

## 3 Different ways of comparing density matrices: Toy examples

In this section, we give three toy examples that illustrate the comparative predictive power of different distances between density matrices. The goals of this section are to (1) emphasize the difference between trace and Schatten distances, and (2) point out how states which are very close under trace distance may be described by very distinct modular Hamiltonians.

**First example. -** In this first toy example, we show that the trace distance can distinguish two states of a large subsystem that any other $n$-distance cannot. Since the Schatten norm has the monotonicity property, this example may not be surprising at all; however, it will help us

to build up the basis for the next two toy examples and for the main arguments. We consider the two $2^\ell \times 2^\ell$ diagonal RDMs:

$$\rho_A = \text{diag}(2^{-\ell}, \cdots, 2^{-\ell}), \qquad \sigma_A = \text{diag}(2^{-\ell+1}, \cdots, 2^{-\ell+1}, 0, \cdots, 0). \tag{17}$$

The RDM $\rho_A$ has $2^\ell$ identical eigenvalues equal to $2^{-\ell}$. The RDM $\sigma_A$ has half of its eigenvalues (i.e. $2^{\ell-1}$) equal to $2^{-\ell+1}$ and the other half are vanishing. Both RDMs are normalized, $\text{tr}_A \rho_A = \text{tr}_A \sigma_A = 1$. The two matrices are very different, and we expect a finite distance between them. It is easy to imagine physical observables being very different in the two cases, for example in a spin system. The trace distance is

$$D(\rho_A, \sigma_A) = \frac{1}{2}, \tag{18}$$

and the normalized $n$-distance (6) is

$$\tilde{D}_n(\rho_A, \sigma_A) = \frac{1}{(1 + 2^{n-1})^{1/n}}, \tag{19}$$

both giving a finite value in the large $\ell$ limit. Conversely, the un-normalized $n$-distance is

$$D_n(\rho_A, \sigma_A) = \frac{1}{2^{(1-\frac{1}{n})\ell + \frac{1}{n}}}. \tag{20}$$

For $n > 1$, $D_n(\rho_A, \sigma_A)$ is exponentially small as $\ell \to \infty$, although the two states are different. We conclude that the trace distance and the normalized distances (6) distinguish these two states for a large subsystem, while the $n$-distance cannot.

**Second example. -** In this second example, we show that the trace distance can distinguish two states of a large subsystem that other normalized $n$-distances cannot. Specifically, we consider:

$$\rho'_A = \frac{1}{2}(\rho_{A,1} + \rho_A), \qquad \sigma'_A = \frac{1}{2}(\rho_{A,1} + \sigma_A), \tag{21}$$

where $\rho_{A,1} = \text{diag}(1, 0, \cdots, 0)$ and $\rho_A, \sigma_A$ are defined in (17). We get a finite trace distance

$$D(\rho'_A, \sigma'_A) = \frac{1}{4}. \tag{22}$$

Conversely, for $n > 1$, the normalized $n$-distance decays exponentially for large $\ell$ as

$$\tilde{D}_n(\rho'_A, \sigma'_A) \approx \frac{1}{2^{(1-\frac{1}{n})\ell + \frac{1}{n}}}. \tag{23}$$

We conclude that the trace distance distinguishes two states of a large subsystem that any other normalized $n$-distance cannot.

**Third example.-** In this last example, we show that two very close RDMs can have different modular Hamiltonians. We consider:

$$\rho_A = \text{diag}(2^{-\ell+1} - 2^{-\ell^2}, \cdots, 2^{-\ell+1} - 2^{-\ell^2}, 2^{-\ell^2}, \cdots, 2^{-\ell^2}),$$
$$\sigma_A = \text{diag}(2^{-\ell+1} - 2^{-2\ell^2}, \cdots, 2^{-\ell+1} - 2^{-2\ell^2}, 2^{-2\ell^2}, \cdots, 2^{-2\ell^2}). \tag{24}$$

Half of the eigenvalues of $\rho_A$ are $2^{-\ell+1} - 2^{-\ell^2}$ and the other half are $2^{-\ell^2}$. Half of the eigenvalues of $\sigma_A$ are $2^{-\ell+1} - 2^{-2\ell^2}$ and the other half are $2^{-2\ell^2}$. These two RDMs are very close in the $\ell \to \infty$ limit, and indeed we have an exponentially small trace distance

$$D(\rho_A, \sigma_A) \approx 2^{-\ell^2 + \ell - 1} \to 0 \quad \text{as } \ell \to \infty. \tag{25}$$

However, they lead to very different modular Hamiltonians

$$-\log \rho_A \approx \mathrm{diag}((\ell-1)\log 2,\cdots,(\ell-1)\log 2,\ell^2 \log 2,\cdots,\ell^2 \log 2),$$

$$-\log \sigma_A \approx \mathrm{diag}((\ell-1)\log 2,\cdots,(\ell-1)\log 2,2\ell^2 \log 2,\cdots,2\ell^2 \log 2). \tag{26}$$

We conclude that, while a direct comparison of modular Hamiltonians and element-by-element entanglement spectra is certainly informative, the trace distance provides a more informative mean of comparing density matrices for large subsystems.

# 4 The XX spin-chain

The XX spin chain (in zero field) is described by the Hamiltonian

$$H_{\mathrm{XX}} = -\frac{1}{4}\sum_{j=1}^{L}\left(\sigma_j^x \sigma_{j+1}^x + \sigma_j^y \sigma_{j+1}^y\right). \tag{27}$$

Its continuum limit is a free massless compact boson theory, with the target space being a unit radius circle, which is a 2D CFT with central charge $c = 1$. The Hamiltonian of the XX spin chain is mapped to that of free fermions by the Jordan-Wigner transformation

$$a_j = \Big(\prod_{i=1}^{j-1}\sigma_j^z\Big)\sigma_j^+, \qquad a_j^\dagger = \Big(\prod_{i=1}^{j-1}\sigma_j^z\Big)\sigma_j^-, \tag{28}$$

with $\sigma_j^\pm = \frac{1}{2}(\sigma_j^x \pm \mathrm{i}\sigma_j^y)$. One can also define the Majorana operators:

$$d_{2j-1} = a_j + a_j^\dagger, \quad d_{2j} = \mathrm{i}(a_j - a_j^\dagger). \tag{29}$$

The exact modular Hamiltonian of an interval $A$ of length $\ell$ takes the form

$$H_A = \sum_{j_1,j_2=1}^{\ell} H_{j_1 j_2} a_{j_1}^\dagger a_{j_2}. \tag{30}$$

The RDM can be constructed from the two-point correlation function matrix [18,59–64] and $H_A$ is related to $C$ as [60,62]

$$C = \frac{1}{1+\mathrm{e}^{H_A}}. \tag{31}$$

In the XX spin chain, the $\ell \times \ell$ correlation matrix $C$ has entries

$$C_{j_1 j_2} = \langle a_{j_1}^\dagger a_{j_2}\rangle = f_{j_2-j_1}. \tag{32}$$

The function $f_j$ takes different forms for different states and geometries. We only consider ground states. For a finite interval $A$ in an infinite chain we have

$$f_j^\infty = \frac{1}{\pi j}\sin\frac{\pi j}{2}, \qquad f_0^\infty = \frac{1}{2}, \tag{33}$$

while for a periodic chain of length $L$ (with $L$ even)

$$f_j^{\mathrm{PBC}} = \frac{\sin\frac{\pi j}{2}}{L\sin\frac{\pi j}{L}}, \qquad f_0^{\mathrm{PBC}} = \frac{1}{2}. \tag{34}$$

Approximate RDM on the lattice [23, 24] can be constructed following the BW theorem [13, 14] and its extensions for CFT [25–34]. For a recent review of BW theorem in quantum field theory, see Ref. [5]. For the ground state of an arbitrary $(d+1)$-dimensional relativistic quantum field theory, the BW theorem states that the modular Hamiltonian of the half-infinite space $A$ (defined by the condition $A = [0, \infty)$) can be written as the partial Lorentz boost generator

$$H_A = 2\pi \int_{x \in A} \mathrm{d}^d x \, x_1 H(x),$$ (35)

where $H(x)$ is the Hamiltonian density of the theory, and the speed of light has been set to unit. For a 2D CFT, the BW theorem can be extended to other geometries [25–34]. For a finite interval $A = [0, \ell]$ on an infinite line in the ground state, the modular Hamiltonian is [26, 29]

$$H_A = 2\pi \int_0^\ell \mathrm{d}x \frac{x(\ell-x)}{\ell} H(x).$$ (36)

For the interval $A = [0, \ell]$ in the ground state of a periodic system of total length $L$, the modular Hamiltonian is [29]

$$H_A = 2\pi \int_0^\ell \mathrm{d}x \frac{\sin \frac{\pi x}{L} \sin \frac{\pi(\ell-x)}{L}}{\frac{\pi}{L} \sin \frac{\pi \ell}{L}} H(x).$$ (37)

We now briefly review how to adapt the continuum formulation to ground states of lattice models [23, 24]. The BW RDM of a given interval is

$$\rho_A^{\mathrm{BW}} = \frac{\mathrm{e}^{-H_A^{\mathrm{BW}}}}{Z_A^{\mathrm{BW}}}, \qquad Z_A^{\mathrm{BW}} = \mathrm{tr}_A \mathrm{e}^{-H_A^{\mathrm{BW}}}.$$ (38)

One can use the modular Hamiltonian in 2D CFT [25–29] to write the BW modular Hamiltonian as

$$H_A^{\mathrm{BW}} = \sum_{j_1, j_2 = 1}^{\ell} H_{j_1 j_2}^{\mathrm{BW}} a_{j_1}^\dagger a_{j_2},$$ (39)

where matrix $H^{\mathrm{BW}}$ has only nearest neighbor non-vanishing entries

$$H_{j,j+1}^{\mathrm{BW}} = H_{j+1,j}^{\mathrm{BW}} = -\pi h_j.$$ (40)

For the interval $A$ in an infinite chain, one has

$$h_j^\infty = \frac{j(\ell-j)}{\ell},$$ (41)

while in a periodic system of length $L$

$$h_j^{\mathrm{PBC}} = \frac{\sin \frac{\pi j}{L} \sin \frac{\pi(\ell-j)}{L}}{\frac{\pi}{L} \sin \frac{\pi \ell}{L}}.$$ (42)

The corresponding matrix of the correlation functions of the BW modular Hamiltonian is

$$C^{\mathrm{BW}} = \frac{1}{1 + \mathrm{e}^{H^{\mathrm{BW}}}}.$$ (43)

## 4.1 One interval embedded in an infinite chain

In this subsection we consider the ground state of an infinite XX chain. By construction, the exact correlation matrix $C$ and the BW one $C^{\mathrm{BW}}$ commute [19]. Hence the corresponding RDM also commute. These commuting RDMs have the same eigenvectors, but may have different eigenvalues that we will use to compute various distances and other related quantities.

### 4.1.1 Trace distance and fidelity

We calculate the trace distance, Schatten $n$-distance, and fidelity between the exact RDM and BW RDM. We exploit the commutativity of the BW RDM with the exact one in our numerical calculations. The $\ell \times \ell$ correlation matrix $C$ defined in (32) with (33) has eigenvalues $\mu_j$, $j = 1, 2, \cdots, \ell$, and in the diagonal basis the $2^\ell \times 2^\ell$ RDM takes the form [61, 63]

$$\rho_A = \bigotimes_{j=1}^{\ell} \begin{pmatrix} \mu_j & \\ & 1-\mu_j \end{pmatrix}. \tag{44}$$

As the correlation matrix $C$ commutes with the BW one $C^{\text{BW}}$, which is defined in (43) with (40) and (41), they can be diagonalized simultaneously. We denote the eigenvalues of the BW correlation matrix $C^{\text{BW}}$ as $\nu_j$, $j = 1, 2, \cdots, \ell$, and, in the same basis as (44), the BW RDM is

$$\rho_A^{\text{BW}} = \bigotimes_{j=1}^{\ell} \begin{pmatrix} \nu_j & \\ & 1-\nu_j \end{pmatrix}. \tag{45}$$

To calculate the trace distance $D(\rho_A, \rho_A^{\text{BW}})$ and general Schatten $n$-distance we need the explicit eigenvalues of the two RDMs. Conversely, thanks to the commutativity of the RDMs $\rho_A$, $\rho_A^{\text{BW}}$, the fidelity is extracted by the simple formula

$$F(\rho_A, \rho_A^{\text{BW}}) = \text{tr}_A \sqrt{\rho_A \rho_A^{\text{BW}}} = \prod_{j=1}^{\ell} \left[ \sqrt{\mu_j \nu_j} + \sqrt{(1-\mu_j)(1-\nu_j)} \right], \tag{46}$$

that does not requires the reconstruction of the spectrum of the RDMs. Similar equations can be found for all Schatten $n_e$-distance with $n_e$ being an even integer

$$D_{n_e}(\rho_A, \rho_A^{\text{BW}}) = \frac{1}{2^{1/n_e}} [\text{tr}_A(\rho_A - \rho_A^{\text{BW}})^{n_e}]^{1/n_e}. \tag{47}$$

For example, the Schatten 2-distance is

$$D_2(\rho_A, \rho_A^{\text{BW}}) = \frac{1}{2^{1/2}} \left\{ \prod_{j=1}^{\ell} \left[ \mu_j^2 + (1-\mu_j)^2 \right] - 2\prod_{j=1}^{\ell} \left[ \mu_j \nu_j + (1-\mu_j)(1-\nu_j) \right] + \prod_{j=1}^{\ell} \left[ \nu_j^2 + (1-\nu_j)^2 \right] \right\}^{1/2}. \tag{48}$$

As a consequence, the calculations of the fidelity and Schatten $n_e$-distances are much more effective because we only need to work with the $\ell$ eigenvalues of the correlation matrix. Hence, we will access subsystem sizes up to $\ell \sim 1000$ (but larger are also possible). In contrast, for the trace distance and Schatten $n_o$-distances we have to reconstruct the $2^\ell$ eigenvalues of the RDMs; for practical reasons we cut off the eigenvalues smaller than $10^{-10}$, but also in this way we can go up at most to $\ell \sim 100$.

The numerical results for the trace distance, Schatten $n$-distance, and fidelity are reported in Fig. 1. From the figure it is evident that all of them decay algebraically as a function of $\ell$, with an exponent that is compatible with $-2$ in all considered cases (in spite of the relatively small $\ell$ that are accessible for the trace distance). Hence, from now on we assume the result $D(\rho_A, \rho_A^{\text{BW}}) \propto \ell^{-2}$, but we stress that our conclusions do not change qualitatively as far as the decay is faster than $\ell^{-1}$.

### 4.1.2 Entanglement entropy

By virtue of the Fannes-Audenaert inequality (10), the behavior of the entanglement entropy is constrained by the trace distance. Since we have found $D(\rho_A, \rho_A^{\text{BW}}) \propto \ell^{-2}$, and since $d_A = 2^\ell$,

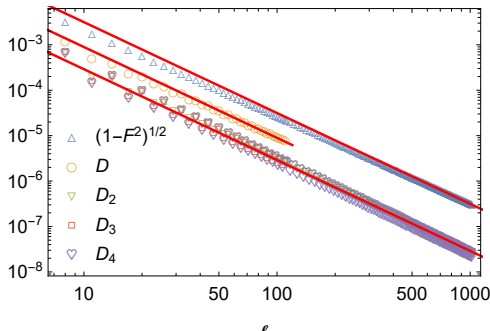

Figure 1: The trace distance $D \equiv D(\rho_A, \rho_A^{\text{BW}})$, Schatten $n$-distance $D_n \equiv D_n(\rho_A, \rho_A^{\text{BW}})$, and fidelity $F \equiv F(\rho_A, \rho_A^{\text{BW}})$ between the BW and the exact RDMs for one interval in an infinite XX chain. The fidelity provides an upper bound for the trace distance, i.e. $D \leq \sqrt{1-F^2}$. The symbols are the spin chain numerical results, and the solid lines are guides for the eyes going as $\ell^{-2}$.

the Fannes-Audenaert inequality characterizes the difference of the entanglement entropies as

$$|S_A - S_A^{\text{BW}}| \lesssim \ell^{-1}. \tag{49}$$

This is consistent with the results in [40,41]. The inequalities for Rényi entropies (11) and (12) do not provide useful bounds; anyhow their absence by no means implies that their differences are not vanishing as $\ell \to \infty$.

We now present numerical computations for the differences of Rényi entropies for several values of $n$, keeping in mind the previous bound from Fannes-Audenaert inequality. We calculate the entanglement entropy and Rényi entropy numerically from correlation matrix following [61,63]. The correlation matrix $C$ has eigenvalues $\mu_j$ with $j = 1, 2, \cdots, \ell$, and the entanglement entropy, Rényi entropy, and RDM moment are

$$S_A = -\sum_{j=1}^{\ell} \left[ \mu_j \log \mu_j + (1-\mu_j) \log(1-\mu_j) \right],$$

$$S_A^{(n)} = -\frac{1}{n-1} \sum_{j=1}^{\ell} \log \left[ \mu_j^n + (1-\mu_j)^n \right],$$

$$\text{tr}_A \rho_A^n = \prod_{j=1}^{\ell} \left[ \mu_j^n + (1-\mu_j)^n \right]. \tag{50}$$

The same formulas are valid for the entanglement entropy, Rényi entropy, and RDM moment for the BW RDM just replacing the eigenvalues $\mu_j$ with those of the BW correlation matrix $C^{\text{BW}}$, that we denoted by $\nu_j$. We emphasize that we need to set a high precision in numerical calculations of the Rényi entropy, especially when the index is in the range $0 < n < 1$. The corresponding results are reported in the left panel of Fig. 2. The entropy difference decays with subsystem size, following approximately the law $|S_A - S_A^{\text{BW}}| \propto \ell^{-2}$, showing that the bound (49) is not tight for the von Neumann entropy. From the figure, it is also evident that also all Rényi entropies with arbitrary $n$ behave exactly in the same fashion, although there is no significant bound in this case.

Now we move to the moments of the RDM for which we have the bounds in Eqs. (14) and (15). Only the one for RDM moments with $n > 1$ is potentially worth to consider:

$$|\text{tr}_A \rho_A^n - \text{tr}_A \rho_{A,\text{BW}}^n| \lesssim \ell^{-2}. \tag{51}$$

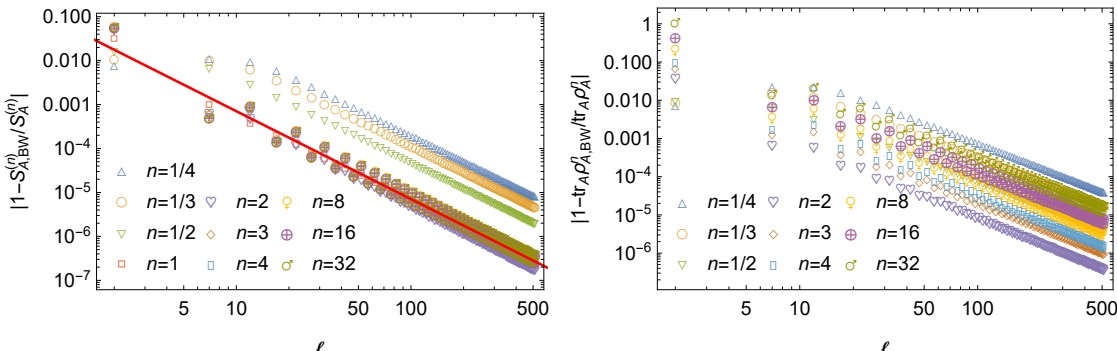

Figure 2: Comparison of the Rényi entropies (left) and RDM moments (right) derived using the BW RDM with the exact ones for an interval of length $\ell$ in the ground state of an infinite XX chain. The symbols are the numerical data and the full line is a guide for the eyes, proportional to $\ell^{-2}$. Evidently, as $\ell \to \infty$, the BW RDM reproduces the exact results.

However as we mentioned in section 2 the moments themselves of the real RDM $\rho_A$ are decaying to zero as $\mathrm{tr}_A \rho_A^n \propto \ell^{-\frac{1}{6}(n-\frac{1}{n})}$. Hence it is more meaningful to look at the ratio for which the bound (15) leads to

$$\left|1 - \frac{\mathrm{tr}_A \rho_{A,\mathrm{BW}}^n}{\mathrm{tr}_A \rho_A^n}\right| \lesssim \ell^{-2+\frac{1}{6}(n-\frac{1}{n})}, \tag{52}$$

which decays to zero as long as $n < 12.08\ldots$. In Fig. 2 we see that all the BW moments converge to the exact ones for any $n$, also rather large. A fit of this decay including all available data points provides exponents that are between 1.8 and 1.9 for all values of $n$ considered. We note however that, since these fits do not consider possible subleading corrections or possible short-distance deviations due to strong-UV effects, these exponents shall be taken with a grain of salt.

### 4.1.3 Two-point fermion correlation function

In this subsection we consider the two-point correlation function of fermion operators, i.e. $\langle a_j^\dagger a_k \rangle$ with $j, k = 1, 2, \cdots, \ell$. These are the building blocks of the correlation matrix $C$ and, by means of Wick theorem, of all local observables within $A$. For these correlations, the inequality (16) with $\mathcal{O} = a_j^\dagger a_k$ constrains their behavior (notice that the maximum singular value $s_{\max}(\mathcal{O})$ is of order one). From the scaling of the trace distance $D(\rho_A, \rho_A^{\mathrm{BW}}) \propto \ell^{-2}$ and the inequality (16), we get the bound for the difference of all observables with $s_{\max}(\mathcal{O})$ of order one:

$$|\langle \mathcal{O} \rangle - \langle \mathcal{O} \rangle_{\mathrm{BW}}| \lesssim \ell^{-2}. \tag{53}$$

When the two fermion operators are at a distance much smaller than $\ell$, $\langle a_j^\dagger a_k \rangle$ is a constant (in $\ell$), and so we conclude

$$|1 - \langle a_j^\dagger a_k \rangle_{\mathrm{BW}}/\langle a_j^\dagger a_k \rangle| \lesssim \ell^{-2}, \qquad \text{if } |j-k| \ll \ell. \tag{54}$$

When the two fermion operators are instead at a distance proportional to $\ell$, we have $\langle a_j^\dagger a_{j+\alpha\ell} \rangle \sim \ell^{-1}$ (cf. Eq. (33)), and so

$$|1 - \langle a_j^\dagger a_k \rangle_{\mathrm{BW}}/\langle a_j^\dagger a_k \rangle| \lesssim \ell^{-1} \qquad \text{if } |j-k| \propto \ell. \tag{55}$$

The comparison between the correlation functions from the BW RDM and the exact ones is shown in Fig. 3. In the first row, we plot two examples at distance 1 and $\ell/2 - 1$, as representative cases of small and large distances respectively. As it is well known, we observe that the BW

correlation function $C^{BW}$ breaks translational invariance. However, such breaking is stronger at small $\ell$ and the symmetry is systematically restored as $\ell$ increases. In the other four panels of Figure 3 we compare the entries of the exact correlation matrix $C$ and the $C^{BW}$, by plotting their relative differences and studying the behavior for large $\ell$. There are two scales that matter: the distance from the boundary of the interval and the distances between the points. Both of them can be either finite or growing with $\ell$ and so we have four possible cases. In the figure we report four examples, one for each possible case. It turns out that the differences for all cases decay like a power law, but the exponent depends on both the relevant scales mentioned above. The numerical results are roughly compatible with the following picture: In the large $\ell$ limit, the difference decays with an exponent in $\ell$ equal (approximately) to $-4$ or $-2$. The decay power is about $-4$, only if both the following conditions are met:

1. the distance of the two operators is finite, and

2. the distances of the two operators from the boundaries of the interval are proportional to $\ell$.

Otherwise, the exponent is approximately $-2$. Let us see how these rules apply to the panel of Fig. 3; only in (c) both conditions are satisfied and the decay is indeed proportional to $\ell^{-4}$ (full line); in all other panels, at most one condition is true (in (d) only (2), in (e) only (1), and in (f) none), hence the decay is proportional to $\ell^{-2}$ (full lines in all panels).

Most of the cases that we considered follow clearly the above picture for the exponent. However, the crossover between the two regimes strongly affects the data for intermediate values of $\ell$, as it is evident from many curves in Fig. 3, but this is not surprising. In Fig. 3 there exist strong dips for some curves at intermediate values of $\ell$, which are due to changes of signs of the differences $1 - C^{BW}_{x,y}/C_{x,y}$. Obviously all these observations about the numerical results are consistent with the bounds (54) and (55).

### 4.1.4 Formation probabilities

In this subsection we study non-local quantities called formation probabilities. The most known of this kind of observables is the emptiness formation probability (EFP) $P_{\pm}$ which has a long history, see [65–77]. $P_+$ ($P_-$) is the probability of having *all* the spins in a block of length $\ell$ pointing up (down) in the $\sigma^z_j$ basis. More generally, a formation probability is defined as $P = \mathrm{tr}_A(\rho_A|\psi\rangle\langle\psi|)$, where $|\psi\rangle$ is a product state of the subsystem $A$ [73, 78]. Using $s_{\max}(|\psi\rangle\langle\psi|) = 1$ for product states and recalling that $D(\rho_A, \rho_A^{BW}) \propto \ell^{-2}$, we get from (16) that any formation probability should satisfy

$$|P - P^{BW}| \lesssim \ell^{-2}, \tag{56}$$

but, as we shall see, this bound is not very meaningful because the FPs themselves are exponentially small.

Let us first analyse the emptiness formation probabilities; they are defined as

$$P_{\pm} = \mathrm{tr}_A(\rho_A \mathcal{O}_{\pm}), \qquad \mathcal{O}_{\pm} = \prod_{j=1}^{\ell} \frac{1 \pm \sigma^z_j}{2}, \tag{57}$$

and can be rewritten as the determinant of an $\ell \times \ell$ matrix $S_{\pm}$ [68, 71]

$$P_{\pm} = \det S_{\pm}, \qquad S^{\pm}_{j_1 j_2} = \frac{1}{2}(\delta_{j_1 j_2} \pm i \langle d_{2j_1-1} d_{2j_2}\rangle). \tag{58}$$

The Majorana modes $d_m$ with $m = 1, 2, \cdots, 2\ell$ are defined in (29). Inversion symmetry of the XX spin chain without transverse field guarantees $P_+ = P_-$. In terms of the correlation matrix

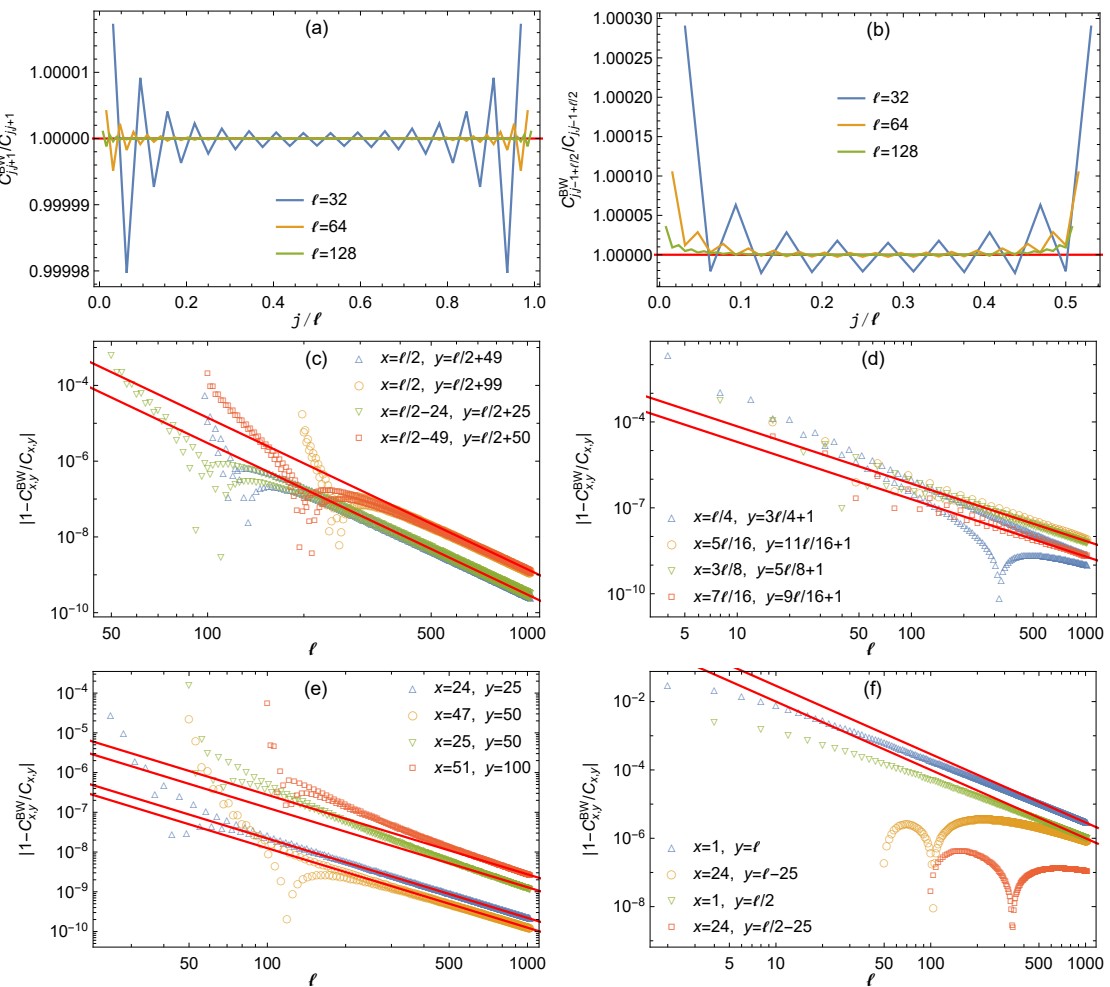

Figure 3: Comparison of exact correlation functions with those coming from the BW RDM for one interval of length $\ell$ in an infinite XX chain. Panels (a) and (b) show that the BW correlations approach the exact ones restoring translational invariance as $\ell \to \infty$. In the panels (c)-to-(f) we report the relative difference of the correlations $|1 - C_{x,y}^{BW}/C_{x,y}|$. In (c), $x - y$ is finite and $x, y \propto \ell$. In (d) $x, y, x - y \propto \ell$. In (e) $x, y$ are both of order 1. In (f) $x$ is $O(1)$, but $y \propto \ell$. In panel (c) the algebraic decay is proportional to $\ell^{-4}$ and in the other three cases to $\ell^{-2}$ (full lines in the various plots).

$C$, we have $S_+ = 1 - C$ and $S_- = C$. The known exact result for the large $\ell$ asymptotics of the EFP of the XX spin-chain in zero field is [68]

$$-\log P_+ = \frac{\log 2}{2} \ell^2 + \frac{1}{4} \log \ell + \cdots, \qquad (59)$$

where the dots are subleading terms in the large $\ell$ limit. Conversely, analyzing the numerical data for the EFP from the BW modular Hamiltonian, we get a very accurate fit with the form[3]

$$-\log P_+^{BW} = \frac{1}{3} \ell^2 + \frac{1}{3} \log \ell + \cdots. \qquad (60)$$

The difference $|\log P_+ - \log P_+^{BW}|$ is shown in figure 4 and grows for large $\ell$ as $\ell^2$, compatible with the forms (59) and (60). This behavior will be discussed at the end of the subsection.

---

[3]This leading order result $\frac{1}{3} \ell^2$ can be obtained analytically by integrating the spectrum of the BW RDM obtained in Ref. [19, 79], as suggested to us by Viktor Eisler.

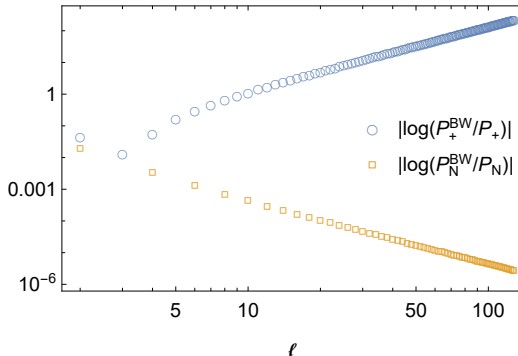

Figure 4: Comparison of the BW EFP and the BW NFP with the exact ones for one interval in the ground state of an infinite XX chain. While the NFP is correctly captured by the BW RDM, the same is not true for the EFP.

We now consider the Néel formation probability (NFP) $P_N$, i.e. the formation probability of the Néel state. This can be written in terms of the matrix $S_+$ in (58) as [73, 78]

$$P_N = (\det S_+)\Big(\text{sdet}\frac{1-S_+}{S_+}\Big), \tag{61}$$

where sdet$(M)$ stands for the determinant of the submatrix of $M$ constructed in such a way that all rows and columns with even (or odd) index are removed. The numerical calculations presented in Ref. [73] suggest the scaling

$$-\log P_N = \frac{\log 2}{2}\ell + \frac{1}{8}\log\ell + \cdots, \tag{62}$$

where the dots stand for subleading terms. We find that the logarithmic NFP derived by using the BW modular Hamiltonian is compatible with this ansatz: in figure 4 we report the difference between the logarithms of the exact and BW NFP, showing that indeed it goes to zero for large $\ell$.

The EFP and NFP themselves are exponentially small as $\ell \to \infty$, and so the inequality (56) are satisfied trivially. The decaying trace distance for large subsystems guarantees the decaying differences of formation probabilities, but the difference of a particular formation probability may not approach each other in $\ell \to \infty$ limit. This is similar to our conclusion in section 3, i.e. that two closed RDMs may have very distinct modular Hamiltonians. For this reason, it is not at all surprising that the EFP is not captured by the BW RDM; quite oppositely, we find remarkable and unexpected that the NFP is well described by BW.

## 4.2 One interval in a finite periodic chain

In this subsection, we move our attention to a finite XX chain of length $L$ with periodic boundary conditions. For conciseness we focus on a subsystem of length $\ell = L/4$, but any other ratio of $\ell/L$ would work the same. In analogy to the study of the infinite system, we first consider the behavior of trace distance, Schatten $n$-distance, and fidelity between the exact RDM $\rho_A$ and the BW RDM, i.e. $\rho_A^{\text{BW}}$, with the modular Hamiltonian defined in (39) with (40) and (42). In our calculations, we rely on the fact that the exact correlation matrix $C$ (32) calculated using the equation (34) commutes with the matrix $H^{\text{BW}}$ (40) calculated by (42), see Ref. [20]. Our results are shown in Fig. 5; the trace distance, the Schatten $n$-distance, and $\sqrt{1-F^2}$ with the fidelity $F$ decay all as approximately $\ell^{-2}$. It is worth mentioning that we find similar results also for open boundary conditions.

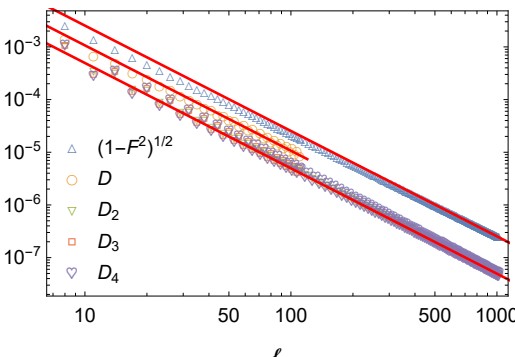

Figure 5: The trace distance, Schatten $n$-distance, and fidelity between the BW and the exact RDMs for one interval in a periodic XX chain. The empty plotmakers are spin chain numerical results, and the solid lines are guide for eyes decaying as $\ell^{-2}$. We fixed the ratio of the length of the interval $\ell$ and the length of the circle $L$ as $\ell = L/4$.

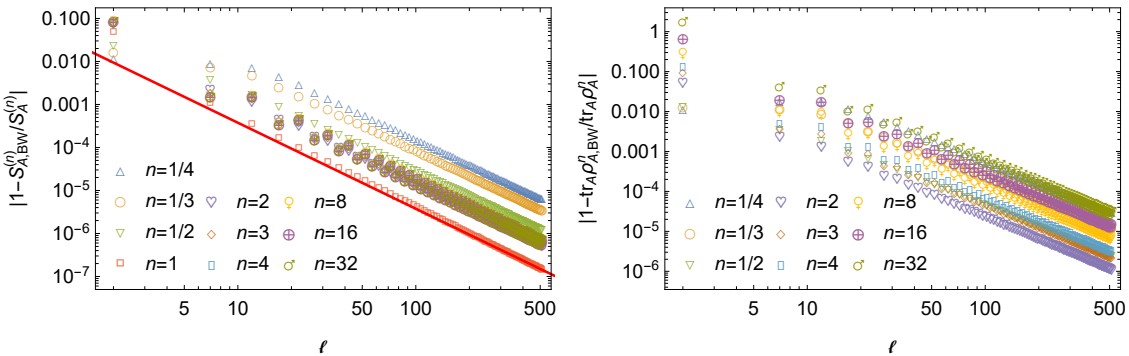

Figure 6: Comparison of the entanglement entropy, Rényi entropies, and RDM moments derived using the BW RDM with the exact ones for one interval in a periodic XX chain with zero magnetic field and for $L = 4\ell$.

We now consider the behavior of the differences of the entanglement entropy, Rényi entropies, and RDM moments in figure 6. We see that approximately $|S_A - S_A^{\mathrm{BW}}| \sim \ell^{-2}$. As in the previous subsection, the BW RDM can reproduce the correct Rényi entropy and RDM moments. We also compare the correlation functions of fermion operators calculated using the BW RDM with the exact ones for one interval on a circle in the ground state of the XX chain. The correlation functions from the BW RDM approach the exact ones for large intervals. The results are very similar to Fig. 3 and we will not show them here.

## 5 The critical Ising chain

The transverse field Ising chain is defined by the Hamiltonian

$$H_{\mathrm{Ising}} = -\frac{1}{2} \sum_{j=1}^{L} \left( \sigma_j^x \sigma_{j+1}^x + \lambda \sigma_j^z \right). \tag{63}$$

Its phase diagram includes a critical point at $\lambda = 1$, on which we focus here. The continuum limit of the critical Ising spin chain is a free massless Majorana fermion, which is a 2D CFT with the central charge $c = \frac{1}{2}$. We recall that, in the massive regime, the entanglement Hamiltonian

of the half-chain in the thermodynamic limit is known to correspond exactly to the lattice version of the BW modular Hamiltonian [15]. The Hamiltonian of the Ising spin chain can be diagonalized exactly by a Jordan-Wigner transformation (28) and moving to Majorana modes (29) as for the XX spin chain. We anticipate that most of the results we obtained for the XX chain will also apply here at the qualitative level; as such, we will keep the discussion of in the form of a short summary, eventually emphasizing differences with respect to the aforementioned case.

Due to the fact that $U(1)$ magnetization conservation is replaced here by a global $\mathbb{Z}_2$ symmetry, we work directly in the Majorana basis $d_m$ with $m = 1, 2, \cdots, 2\ell$ defined by Eq. (29). The exact modular Hamiltonian of a length $\ell$ interval takes the form

$$H_A = \frac{1}{2} \sum_{m_1, m_2 = 1}^{2\ell} W_{m_1 m_2} d_{m_1} d_{m_2}. \tag{64}$$

The correlation matrix $\Gamma$ of Majorana operators is defined as

$$\Gamma_{m_1 m_2} = \langle d_{m_1} d_{m_2} \rangle - \delta_{m_1 m_2}. \tag{65}$$

In the ground state of the critical Ising spin chain, the non-vanishing entries of the correlation matrix satisfy

$$\Gamma_{2j_1-1, 2j_2} = -\Gamma_{2j_2, 2j_1-1} = g_{j_2-j_1}. \tag{66}$$

For the interval embedded in the ground state of an infinite chain we have

$$g_j^\infty = -\frac{i}{\pi} \frac{1}{j + \frac{1}{2}}, \tag{67}$$

while on a periodic system of length $L$

$$g_j^{\text{PBC}} = -\frac{i}{L \sin[(j + \frac{1}{2})\frac{\pi}{L}]}. \tag{68}$$

The matrices $W$ and $\Gamma$ are related as [80]

$$\Gamma = \tanh W. \tag{69}$$

The BW modular Hamiltonian of the critical Ising spin chain is

$$H_A^{\text{BW}} = \frac{1}{2} \sum_{m_1, m_2 = 1}^{2\ell} W_{m_1 m_2}^{\text{BW}} d_{m_1} d_{m_2}, \tag{70}$$

with the nonvanishing entries of the antisymmetric matrix

$$
\begin{aligned}
W_{2j-1, 2j}^{\text{BW}} &= -W_{2j, 2j-1}^{\text{BW}} = -\pi i h_{j-\frac{1}{2}}, \quad j = 1, 2, \cdots, \ell, \\
W_{2j, 2j+1}^{\text{BW}} &= -W_{2j+1, 2j}^{\text{BW}} = -\pi i h_j, \quad j = 1, 2, \cdots, \ell - 1.
\end{aligned}
\tag{71}
$$

The function $h_j$ is defined as (41) or (42), depending on the interval of interest. The corresponding correlation matrix of the BW modular Hamiltonian is

$$\Gamma^{\text{BW}} = \tanh W^{\text{BW}}. \tag{72}$$

## 5.1 One interval in an infinite chain

In this subsection we consider the ground state of an infinite chain. The BW modular Hamiltonian is defined as (70) with (71) and (41).

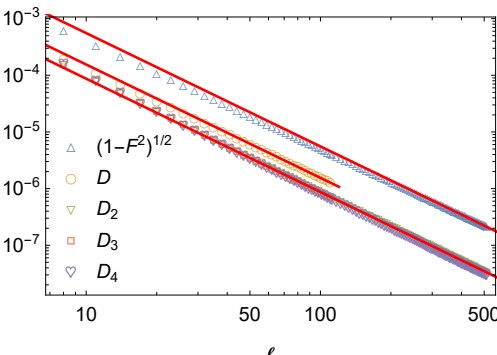

Figure 7: The trace distance, Schatten $n$-distance, and fidelity of the BW and the exact RDMs for one interval on an infinite chain in the ground state of the critical Ising spin chain. The empty plotmakers are spin chain numerical results. The solid lines are guide to the eyes going like $\ell^{-2}$.

### 5.1.1 Trace distance and fidelity

The exact correlation matrix $\Gamma$ (65) with (66) and (67) and the matrix $W^{\text{BW}}$ (71) with (41) commute. The exact correlation matrix $\Gamma$ has eigenvalues $\pm\gamma_j$ with $j = 1, 2, \cdots, \ell$, and the BW correlation matrix $\Gamma^{\text{BW}}$ (72) can be diagonalized under the same basis with the eigenvalues $\pm\delta_j$ with $j = 1, 2, \cdots, \ell$. In the same basis, the exact and the BW RDM can be written respectively in the diagonal forms

$$\rho_A = \bigotimes_{j=1}^{\ell} \begin{pmatrix} \frac{1+\gamma_j}{2} & \\ & \frac{1-\gamma_j}{2} \end{pmatrix}, \qquad \rho_A^{\text{BW}} = \bigotimes_{j=1}^{\ell} \begin{pmatrix} \frac{1+\delta_j}{2} & \\ & \frac{1-\delta_j}{2} \end{pmatrix}. \tag{73}$$

To calculate the trace distance and Schatten $n_o$-distance with $n_o$ being an odd integer we need the explicit eigenvalues of the RDMs, while we have simpler formulas to calculate the Schatten $n_e$-distance with $n_e$ being an even integer and fidelity. As for the XX chain, we discard the eigenvalues of the RDMs smaller than $10^{-10}$, and this allow us to calculate the trace distance and Schatten $n_o$-distance up to $\ell \sim 100$. However, this limitation does not apply to fidelity and Schatten $n_e$-distance for which we go up to $\ell \sim 500$. The trace distance, Schatten distance, and the fidelity of the exact RDM and the BW RDM are shown in the figure 7. Similarly to the XX chain, the BW RDM becomes an increasingly better approximation of the exact RDM as the subsystem size $\ell \to \infty$. By numerical fitting we get that the trace distance, Schatten distance and $\sqrt{1-F^2}$ with $F$ being the fidelity all decay approximately as $\ell^{-2}$.

### 5.1.2 Entanglement entropy

In this subsection we study the behavior of the differences of the entanglement entropy, Rényi entropies, and the RDM moments in figure 8. We see that approximately $|S_A - S_A^{\text{BW}}| \propto \ell^{-2}$. It is consistent with the results in [41]. Again, similarly to the XX case, the BW RDM reproduces all the exact Rényi entropies and RDM moments. The difference of Rényi entropies decay as $\ell^{-2}$, while the difference of the moments decay with an exponent slightly smaller than 2, in full analogy with the XX chain.

We mention that we tested also the scaling of the correlation functions. All results are identical to the ones of the XX chain in Figure 3 and so we do not discuss them here.

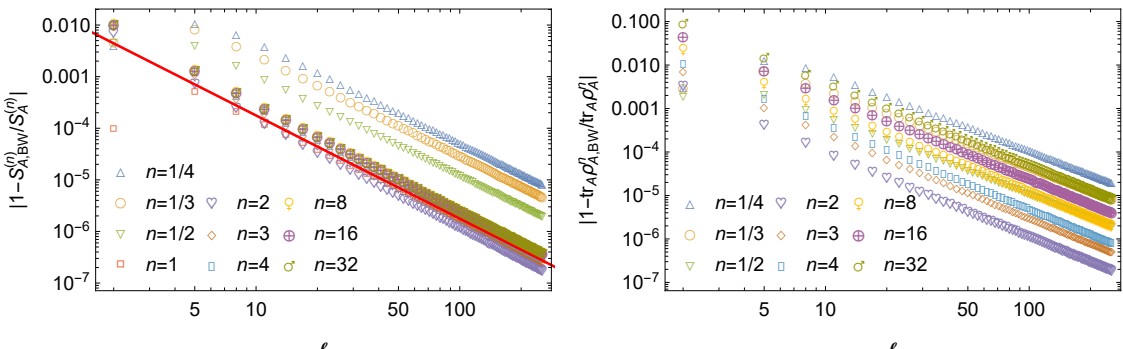

Figure 8: Comparison of the entanglement entropy, Rényi entropies, and RDM moments derived using the BW RDM with the exact ones for one interval on an infinite chain in the ground state of the critical Ising chain. The full line on the left is proportional to $\ell^{-2}$.

### 5.1.3 Formation probabilities

In the critical Ising chain, the EFPs $P_\pm$ for all the spins up and all the spins down in the basis of $\sigma_j^z$ are different. For the critical Ising spin chain, there are the exact results [71]

$$-\log P_\pm = \left(\log 2 \mp \frac{2G}{\pi}\right)\ell + \frac{1}{16}\log \ell + \cdots, \qquad (74)$$

where $G$ is the Catalan's constant. Note that $P_+ > P_-$, i.e. the configuration with all the spins up (in the transverse direction) is preferred to the one with spin down, as obvious from energetic arguments. In terms of the correlation matrix $\Gamma$ (65) with (66) and (67), we have [68, 71]

$$P_\pm = \det S_\pm, \qquad S_{j_1 j_2}^\pm = \frac{1}{2}(\delta_{j_1 j_2} \pm i\Gamma_{2j_1-1,2j_2}). \qquad (75)$$

The same formula works also for the BW correlation matrix $\Gamma^{\text{BW}}$ (72) with (71) and (41) to get the EFP $P_\pm^{BW}$. The numerical results are shown in figure 9. Unlike the case in the XX spin chain, the BW modular Hamiltonian in the Ising spin chain perfectly reproduces the EFPs.

In the critical Ising spin chain numerical calculations suggest that for the NFP we have [73]:

$$-\log P_{\text{N}} \approx 0.985\ell + \frac{1}{16}\log \ell + \cdots. \qquad (76)$$

We find NFP coming from the BW modular Hamiltonian also matches this result when the subsystem size is large, as shown in figure 9.

### 5.2 One interval in a finite periodic chain

In this subsection we consider the ground state of a periodic system of length $L = 4\ell$. The exact correlation matrix $\Gamma$ (65) with (66) and (68) commute with $W^{\text{BW}}$ (71) with (42). The trace distance, Schatten distance and fidelity of the exact RDM and BW RDM are shown in figure 10. The trace distance, Schatten distance and $\sqrt{1-F^2}$ with $F$ being the fidelity all decay approximately as $\ell^{-2}$.

We show the behavior of the differences of the entanglement entropy, Rényi entropies, and RDM moments in Figure 11. We see that approximately $|S_A - S_A^{\text{BW}}| \propto \ell^{-2}$. The BW RDM reproduces the exact entanglement entropy, Rényi entropies and RDM moments for large subsystem sizes.

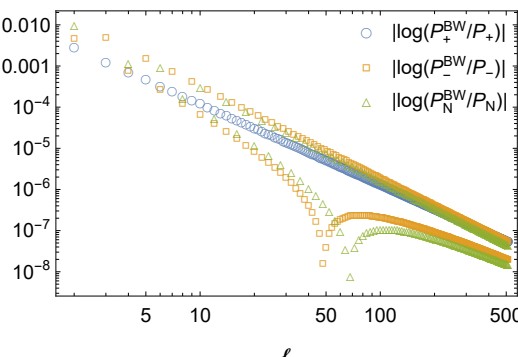

Figure 9: Comparison of the logarithmic EFPs and logarithmic NFP coming from the BW RDM with the exact ones for one interval on an infinite chain in the ground state of the critical Ising chain. Notice that $P_-$ and $P_N$ displays even-odd effects.

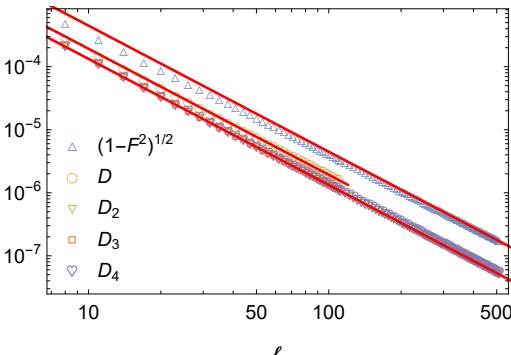

Figure 10: The trace distance, Schatten $n$-distance, and fidelity of the BW and exact RDMs for one interval in the ground state of a periodic critical Ising spin chain. The empty plotmakers are spin chain numerical results, and the solid lines are decaying as $\ell^{-2}$. We have fixed the ratio of the length of the interval $\ell$ and the length of the circle $L$ as $\ell = L/4$.

## 6 Conclusion

We compared the reduced density matrices coming from the discretization of the BW modular Hamiltonian and the exact ones for the ground state of the XX chain with a zero magnetic field and the critical transverse field Ising chain. In order to provide a meaningful metric comparison between density matrices, we have considered the trace distance between the exact RDM and the BW RDM as our main figure of merit. We found that the trace distance between BW RDMs and the exact ones goes to zero approximately as $\ell^{-2}$ when the length of the interval $\ell$ goes to infinity. The behavior of the trace distance gives strong constraints on the entanglement entropy and correlation functions. The differences of the entanglement entropy and correlation functions also go to zero as $\ell \to \infty$.

We have found that the Rényi entropies and moments of any order are all well reproduced by the BW RDM. Their differences decay algebraically for large $\ell$, although in most cases the known sharp bounds provides no constraint at all. It is not known yet what special properties of the ground state of critical system enforce such peculiar scaling, but it is clearly worth investigating because it is related to the working accuracy of the BW RDM. However, we found that the most peculiar behavior is displayed by the formation probabilities. Indeed, in the ground state they decay exponentially fast with the length of the interval and so, a priori, we

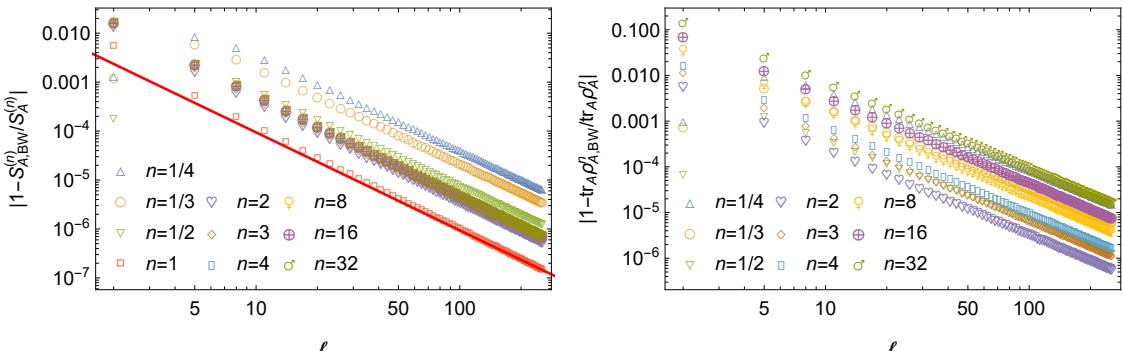

Figure 11: Comparison of the entanglement entropy, Rényi entropies, and RDM moments coming from the BW RDM with the exact ones for one interval on a circle in the ground state of the critical Ising chain.

would not have expected to be well reproduced by the BW approximation. Instead, only the emptiness formation probability of the XX chain is not captured by the BW RDM, while the Néel formation probability in XX and all the studied ones in Ising are, surprisingly, captured by this approximation. We do not know whether and how this could be connected with the Gaussian vanishing for large $\ell$ of the EFP in the XX chain, while all the others are simple exponentials.

# Acknowledgements

We thank V. Eisler for helpful discussions and reading the manuscript. MD and MAR thank G. Giudici and T. Mendes-Santos for discussions and collaborations on related topics. MAR also thanks ICTP and SISSA for hospitality during the initial and final stages of this work.

**Funding information** PC and JZ acknowledge support from ERC under Consolidator grant number 771536 (NEMO). MD acknowledges support ERC under grant number 758329 (AGEnTh), and the European Union's Horizon 2020 research and innovation programme under grant agreement No 817482 (Pasquans). MAR thanks CNPq and FAPERJ (grant number 210.354/2018) for partial support.

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
