# Peer review of "Lattice Bisognano-Wichmann modular Hamiltonian in critical quantum spin chains"

_SciPost Physics Core, doi:SciPost Phys. Core 2, 007 (2020)_

## Round 1 · Referee Report · Anonymous (Referee 1) · 2020-3-27

Strengths

1: very well written, clear, easy to follow.
2: very pedagogical, step-by-step approach to the raised issues
3: clarifying in a quantitative way on a number of explicit examples the quantitative relevance of the BW construction for investigating entanglement effects in lattice field theories at large subsystem size.
4: challenging results: the algebraic scaling (exponent-2) of the convergence; the remarkable convergence of the NFP. This clearly open new and exciting perspectives.

Weaknesses

An introduction to BW theorem and its uses in this paper is really needed.

Report

The authors address the question of comparing a number of observables relevant for entanglement in 1-d spin chains , using the exact modular Hamiltonian (in several of the few cases where it is available) versus an approximate Hamiltonian, obtained by applying an extension to lattice field theories of the Bisognano-Wichmann theorem of constructive field theory, developed by several of the authors in previous papers. Such a construction is always available, and it is indeed an important issue to check its validity .

The authors propose a characterization of a relevant "distance" between density matrices for lattice field theories resp. exact; and obtained by the BW theorem . They then compare the behaviour of this distance (and others quantities such as Renyi entropies) when the size of the subsystem goes to infinity. They confront results for physical observables, correction functions and formation probabilities. A surprising scaling of the distance itself, and some of the observables , is identified, and an unexpected convergence of some FP is also identified.

For specific comments see Strengths/Weaknesses.

Requested changes

1: explain more precisely what the BW formalism is, e.g. when introducing 4.10. A brief summary of the derivation in the previous paper arXiv 1807.01322 would in this respect be useful.
2: an explanation of the qualitative difference between the computation of fidelity and even-Schatten distances on one hand; and trace and generic odd-Schatten distance on the other hand, around formulae 4.16-4.18; would be appreciated.
3: I am slightly confused by the conclusion of section 4.1.2 "close to 2 but likely different from 2" ? Is there a possibility to estimate more precisely the decay exponent from numerical results ?
4: The log-difference between correlation functions in Fig. 3 exhibits strong dips with sometimes "oscillatory-like" behavior at intermediate scales for l. Do the authors have some qualitative or even quantitative understanding of this particular behaviour ?

---

## Round 1 · Referee Report · Anonymous (Referee 2) · 2020-4-12

Report

Lattice Bisognano-Wichmann modular Hamiltonian in critical quantum spin chains

by Jiaju Zhang, Pasquale Calabrese, Marcello Dalmonte, M. A. Rajabpour

The present paper concerns the reduced-density-matrix description of a large subsystem of two critical quantum chains.
More precisely,
the authors consider the spin-1/2 XX chain in zero field and the spin-1/2 Ising chain in the critical transverse field;
for both exactly diagonalizable systems the reduced density matrix ρA and the modular Hamiltonian HA can be constructed exactly.
Then, for these critical chains,
the authors construct the (approximate) Bisognano-Wichmann modular Hamiltonian HBWA and the corresponding reduced density matrix ρBWA.
The goal of their study is
to examine various distances between the reduced density matrices ρBWA and ρA
(and some related quantities including formation probabilities)
when the size of the subsystem A of length l becomes large, i.e., l.
This analysis illustrates the precision of the Bisognano-Wichmann modular Hamiltonian approach
for calculation of various measures of quantum entanglement in the critical quantum spin chains at hand.
The differences for most quantities
(calculated exactly and with the help of the approximate Bisognano-Wichmann modular Hamiltonian)
decay algebraically as a function of l.
The only exception is the emptiness formation probability of the XX chain:
It cannot be captured by the the Bisognano-Wichmann modular Hamiltonian approach.

In my opinion,
the authors present a set of useful results which satisfy acceptance criteria and deserve publication;
the paper is clearly written;
it should be useful for the community studying entanglement questions for quantum many-body systems.

---

## Round 2 · Referee Report · Anonymous (Referee 2) · 2020-4-23

Report

I have checked the response and the revised version of the paper; I am completely satisfied with the present version and recommend to publish the paper.

---

## Round 2 · Referee Report · Anonymous (Referee 1) · 2020-4-23

Report

The few questions and suggestions in the referee's report have been suitably answered. The paper is now acceptable for publication as it stands

---

## Round 2 · Author Response

We thank both referees for positive judgment on our paper. We took into account the comments by the first referee and made a few minor changes that we summarize below.

---

## Round 2 · List of Changes

Answer 1)
We added a few sentences and equations on page 8 and explained the BW theorem.
Answer 2)
The computation of fidelity and even-Schatten distances do not require the diagonalization of the difference between reduced density matrices. They are just traces of sums of products of Gaussian operators that can be calculated using the standard Gaussian algebra. Conversely, to the best of our knowledge, the trace distance and odd-Schatten ones can be calculated only by explicitly diagonalizing the difference between reduced density matrices. We added equations (46) and (47) on page 9 to make this clear.
Answer 3)
We added a sentence (after equation (52)) about a fit for these exponents that should be taken very carefully.
Answer 4)
The dips are due to the change in sign of the differences between BW and real correlations. We do not believe there is any relevant physics behind this change of sign. In the new version, we added a sentence (page 13) stressing the existence of the dip.

---

## Editorial Decision

published